# MM-LDM: Multi-Modal Latent Diffusion Model for Sounding Video Generation

## ABSTRACT

Sounding Video Generation (SVG) is an audio-video joint generation task challenged by high-dimensional signal spaces, distinct data formats, and different patterns of content information. To address these issues, we introduce a novel multi-modal latent diffusion model (MM-LDM) [1] for the SVG task. We first unify the representation of audio and video data by converting them into a single or a couple of images. Then, we introduce a hierarchical multi-modal autoencoder that constructs a low-level perceptual latent space for each modality and a shared high-level semantic feature space. The former space is perceptually equivalent to the raw signal space of each modality but drastically reduces signal dimensions. The latter space serves to bridge the information gap between modalities and provides more insightful cross-modal guidance. Our proposed method achieves new state-of-the-art results with significant quality and efficiency gains. Specifically, our method achieves a comprehensive improvement on all evaluation metrics and a faster training and sampling speed on Landscape and AIST++ datasets. Moreover, we explore its performance on open-domain sounding video generation, long sounding video generation, audio continuation, video continuation, and conditional single-modal generation tasks for a comprehensive evaluation, where our MM-LDM demonstrates exciting adaptability and generalization ability.

## CCS CONCEPTS

• **Computing methodologies** → *Hierarchical representations*; Neural networks; **Computer vision tasks**.

## KEYWORDS

Multi-modal Generation, Sounding Video Generation, Latent Diffusion Model, Audio Generation, Video Generation

## 1 INTRODUCTION

Sound Video Generation (SVG) is an emerging task in the field of multi-modal generation, which aims to integrate auditory and visual signals for audio-video joint generation [18, 32, 33]. The integrated sounding videos closely simulate real-life video formats, providing immersive audiovisual narratives [2, 3, 36]. The potential applications of SVG span multiple fields, including film production,

game development, virtual reality, etc, making it an area worth exploring in depth.

Compared with single-modal generation, SVG is a more challenging task since it requires a deep understanding of the complex interactions between auditory and visual content [24]. Specifically, three primary challenges hinder the progress of SVG. **First**, both video and audio are high-dimensional data, making it difficult to achieve realistic generation of both modalities, especially when computational resources are constrained. **Second**, video and audio data have distinct dimensions and representations, necessitating specified designs to obtain a unified architecture. In particular, audios are 1D continuous auditory signals with a single amplitude channel, which focus on temporal information, whereas videos are 3D visual signals with RGB color channels, which involve both spatial and temporal information. Previous work [24] unifies video and audio generation by coupling two modal-specific models using random-shift based attention layers. However, the calculation of the attention is based on a small cross-modal attention window, limiting cross-modal communication and thus obtaining suboptimal cross-modal consistency. **Third**, patterns of content information conveyed by videos and audio are distinct, which significantly exacerbates the difficulty in obtaining cross-modal consistency. Videos record dense visual dynamics that evolve over time, while audios record sound waves made by both various visible and invisible sources. Given that messages conveyed by these modalities can vary dramatically, the generation of consistent audios and videos requires model to understand high-level semantic information of both modalities, e.g., identify which kind of sound can be made given the video scene, thus posing a formidable challenge for current generative models.

To address the above challenges, we propose a novel Multi-Modal Latent Diffusion Model (MM-LDM) for SVG. Firstly, to reduce the dimension of video and audio data, we introduce a multi-modal auto-encoder that constructs two perceptual latent spaces, which are modal-specific, low-dimensional, and perceptually equivalent to the raw signal spaces. By modeling SVG within the perceptual latent spaces, we significantly reduce the computation burden and improve the generation efficiency.

Secondly, to facilitate the design of a unified framework that processes both modalities, we unify the dimensions and representations of video and audio data by encoding audio signals into an audio image and treating video signals as a sequence of images (i.e., video frames). Such unified representation allows our multi-model auto-encoder to utilize a pretrained image diffusion model as the signal decoder and share its parameters for both modalities. Moreover, we can explicitly model the temporal alignment of two modalities within the perceptual latent space since the perceptual latents are perceptually equivalent to raw signals, thus enabling better temporal alignment, superior synthesis quality, and improved coherence throughout the generated sample.

---

[1]Our codes have been partly released in https://anonymous.4open.science/r/MM-LDM/ and will be completely released soon.

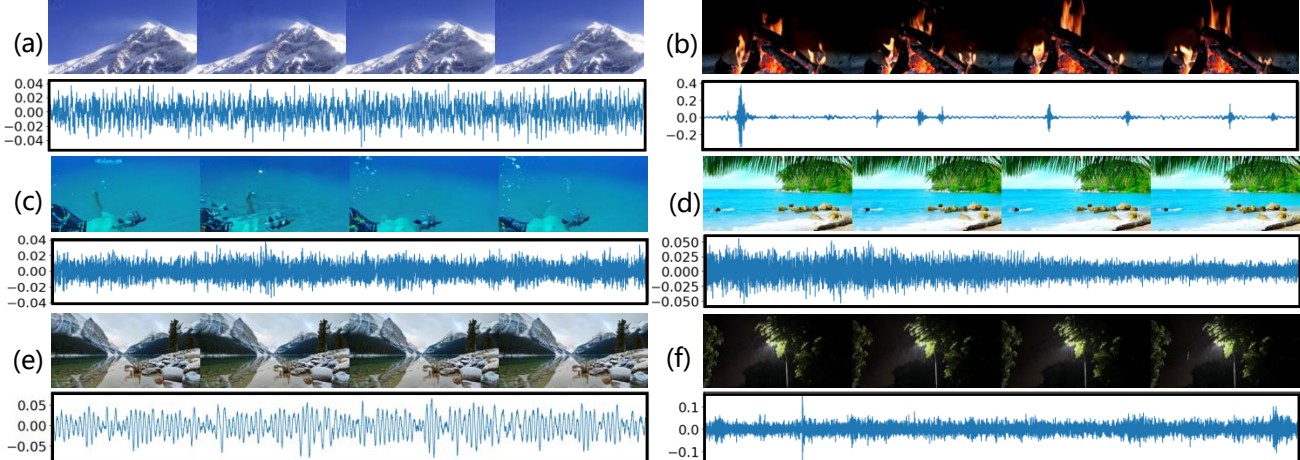

Figure 1: Sounding videos generated by our MM-LDM on the Landscape dataset [15]. We can observe vivid scenes like (a) mountain, (c) diving man, (e) lake, and so on. Matched audios are given like the sound of (b) wood burning, (d) sea wave, (f) raining, and so on. All presented audios (in this paper) can be played in Adobe Acrobat by clicking corresponding wave figures. More playable sounding video samples can be found in https://anonymouss765.github.io/MM-LDM.

Thirdly, to bridge the gap between content information conveyed by two modalities, we strengthen the modeling of multi-modal correlations in both decoding and generation processes. During the decoding process, we derive a shared semantic space based on the perceptual latents to provide cross-modal guidance. Specifically, two projectors are used to map the perceptual latents into the shared semantic space. Moreover, we introduce two multi-modal semantic losses including a classification loss and a contrastive loss to optimize the high-level semantic features during training. During the generation process, we model both single-modal and cross-modal correlations using full attention. In particular, a full self-attention based Transformer [20] is used as the backbone of the diffusion model. It takes rasterized and concatenated audio and video perceptual latents as inputs, building both single-modal and cross-modal correlations using multiple self-attention layers.

To the best of our knowledge, MM-LDM is the first latent diffusion model for the SVG task, which requires audio-video joint generation. We introduce multiple specialized designs tailored for this multi-modal generation task as specified above. Furthermore, our proposed method exhibits remarkable adaptability when extended to various other multi-modal generation tasks, including open-domain sounding video generation, audio-to-video generation, video-to-audio generation, long sounding video generation, audio continuation, video continuation, and so on. As shown in Fig. 1, MM-LDM can synthesize high-resolution ($256^2$) sounding videos with vivid objects, realistic scenes, coherent motions, and aligned audio-video content. We conduct extensive experiments on the Landscape [15], AIST++ [16], and AudioSet [6] datasets, achieving new state-of-the-art generation performance with significant visual and auditory gains. For example, on the AIST++ dataset with $256^2$ spatial resolution, our MM-LDM outperforms MM-Diffusion by 114.6 FVD, 21.2 KVD, and 2.1 FAD. We also reduce substantial computational complexity, achieving a 10x faster sampling speed and allowing a larger sampling batch size. Our contributions can be summarized as follows:

- We propose a novel multi-modal latent diffusion model that establishes low-dimensional audio and video latent spaces for SVG, which are perceptually equivalent to the original signal spaces but significantly reduce the computational complexity.
- We derive a shared high-level semantic feature space from the low-level perceptual latent spaces to provide cross-modal guidance and bridge the information gap between audio-video content during the decoding process.
- We introduce multiple cross-modal losses to improve cross-modal consistency and optimize the semantic feature space, including an audio-video adversarial loss, an audio-video contrastive loss, and a classification loss.
- We perform a comprehensive evaluation and extend our method to multiple other generation tasks, which demonstrates the effectiveness, efficiency, and adaptability of our proposed method, where we achieve new state-of-the-art results with significant quality and efficiency gains.

## 2 RELATED WORK

***Sounding Video Generation***. In recent years, several works have been proposed to explore the challenging task of SVG. Based on generative adversarial networks, HMMD [14] introduces multiple discriminators to guide the generator in producing sounding videos with aligned audio-video content. Based on sequential generative models, UVG [18] introduces a multi-modal tokenizer to encode different modal signals into discrete tokens, and employs a Transformer-based generator for SVG. Although these works have tackled the SVG task to some extent, their performances are far from expectations. Drawing inspiration from the success of diffusion models [10, 29], many works have explored multi-modal diffusion models for audio-video generation [24, 33, 36], while most methods focus on generating one modality based on another [19, 30, 33, 36].

Recently, MM-Diffusion [24] stands out as the pioneer in simultaneously synthesizing both modalities. This approach introduced two-couple denoising diffusion models to gradually generate aligned audio-video signals from pure noises. To align generated audio and video content, MM-Diffusion introduces a random-shift based attention for cross-modal communication. However, this method suffers from a huge computational burden since it addresses SVG in the signal space, and uses a limited attention window size (typically no more than 8), resulting in sub-optimal cross-modal consistency. Contrastively, we establish low-level latent spaces to reduce computational complexity and a high-level latent space to provide cross-modal guidance. Our method significantly outperforms MM-Diffusion in terms of both generation efficiency and quality.

**Latent Diffusion Model.** Given that raw signal spaces for image, audio, and video modalities are of high dimensions, extensive efforts have been devoted to modeling their generation using latent diffusion models [1, 7, 23, 34]. For the image modality, LDM [23] is devised to construct a perceptual latent space for images. This approach introduces a KL-VAE to encode images into image latents and utilizes a latent diffusion model for text-to-image generation within the latent space. For the audio modality, AudioLDM [17] is introduced to facilitate text-to-audio generation in a 1D latent space. In particular, it utilizes a large text-audio pretrained model CLAP [31] for extracting text and audio latent features. For the video modality, VideoLDM [1] is proposed to extend the LDM [23] to high-resolution video generation. It introduces a temporal dimension into the LDM, and only optimizes these temporal layers while maintaining fixed, pretrained spatial layers. Most previous latent diffusion models were primarily designed to encode a single modality, thus incorporating a single perceptual latent space. Unlike prior works, our MM-LDM introduces a hierarchical multi-modal autoencoder that establishes both low-level perceptual latent spaces and a high-level semantic feature space. The former, similar to prior latent diffusion models, is perceptually equivalent to raw modality signals and is important for reducing computational complexity. Differently, the latter, a distinctive space of our MM-LDM, is built based on the perceptual latent space and is specifically devised to enhance the consistency of generated audio-video content. Such hierarchical spaces contribute to the enhanced ability to address the challenging multi-modal generation task.

## 3 METHOD

In this section, we present our multi-modal latent diffusion model (MM-LDM) in detail. This approach consists of two main components: a hierarchical multi-modal autoencoder designed for compressing video and audio signals, and a multi-modal latent diffusion model for modeling SVG within latent spaces. An overview of MM-LDM is shown in Fig. 2

### 3.1 Hierarchical Multi-Modal Autoencoder

As shown in Fig. 2 and Fig. 3, the autoencoder is composed of two modal-specific encoders, two signal decoders with shared parameters, two projectors mapping from each perceptual latent space to the shared semantic feature space, and two heads for classification and contrastive learning, respectively. Notably, our multi-modal autoencoder establishes two hierarchical feature spaces: a perceptual

latent space that aligns with raw signals, and a semantic feature space derived from the perceptual space for bridging the information gap between audio and video modalities.

**Unifying Representation of Video and Audio Signals.** We employ the raw video signals $v$ and transformed audio images $a$ to be our inputs. Video $v \in \mathbb{R}^{F \times 3 \times H \times W}$ can be viewed as a sequence of 2D images (i.e. video frames), where $F$, 3, $H$, and $W$ are video length, dimension, height, and width, respectively. Given that raw audio signals are 1D-long continuous data, we transform raw audio signals into 2D audio images to reduce computation complexity and unify the representation of audio and video inputs. In particular, given a raw audio clip, we first obtain its Mel Spectrogram with values normalized, which is denoted as $a_{raw} \in \mathbb{R}^{D \times T}$, where $D$ represents the number of audio channels and $T$ is the temporal dimension. Notably, $a_{raw}$ can be transformed to raw 1D audio signals using pretrained HiFiGAN [12]. Then, we treat $a_{raw}$ as a grayscale image and convert it into an RGB image using the PIL Python toolkit. Finally, we resize the image to the same spatial resolution as the video input, obtaining an audio image $a \in \mathbb{R}^{3 \times H \times W}$.

**Modal-specific Perceptual Latent Spaces.** Given a pair of audio and video inputs, we employ a pretrained KL-VAE [23] to downsample the video frames and audio images by a factor of $f$. Then, as depicted in Fig. 3(a), we introduce an audio encoder to compress the audio image into an audio perceptual latent $z_a \in \mathbb{R}^{C \times H_a \times W_a}$, which further downsamples the audio image by a factor of $f_a$, where $C$ is the number of channels, $H_a$ and $W_a$ denotes $\frac{H}{f \times f_a}$ and $\frac{W}{f \times f_a}$, respectively. Similarly, the video encoder compresses video frames into $z_v \in \mathbb{R}^{C \times H_v \times W_v}$. Notably, the video encoder requires additional temporal layers to capture temporal correlations between video frames. Moreover, pixel distributions of audio images and video frames differ greatly since audio images are typical images in physics that depict wave features while video frames record real-life object appearances. Based on the above insights, parameters of the audio and video encoder are not shared and their perceptual latent spaces are modal-specific. More details about the encoder structures are presented in the supplementary material.

**A Shared High-level Semantic Space.** For audio-video cross-modal consistency, auxiliary information is required in the decoding process of audio and video latent features. In our experiments, we observe a significant performance drop when we directly use one perceptual latent as condition input to provide cross-modal information when decoding another perceptual latent. This performance drop reveals that it is hard for the signal decoder to extract useful cross-modal information from perceptual latent features. This can be mainly attributed to the fact that perceptual latents are dense representations of low-level information, thus presenting challenges for the decoder to comprehend.

To provide the decoder with useful cross-modal guidance, specialized modules are required to extract high-level information from perceptual latents and narrow the gap between video and audio features. To this end, as depicted in Fig. 3(a), we introduce an audio projector and a video projector that establish a shared high-level semantic space based on the low-level perceptual latents. In particular, the audio and video projectors extract semantic audio and video features $s_a$ and $s_v$ from their perceptual latents. To ensure

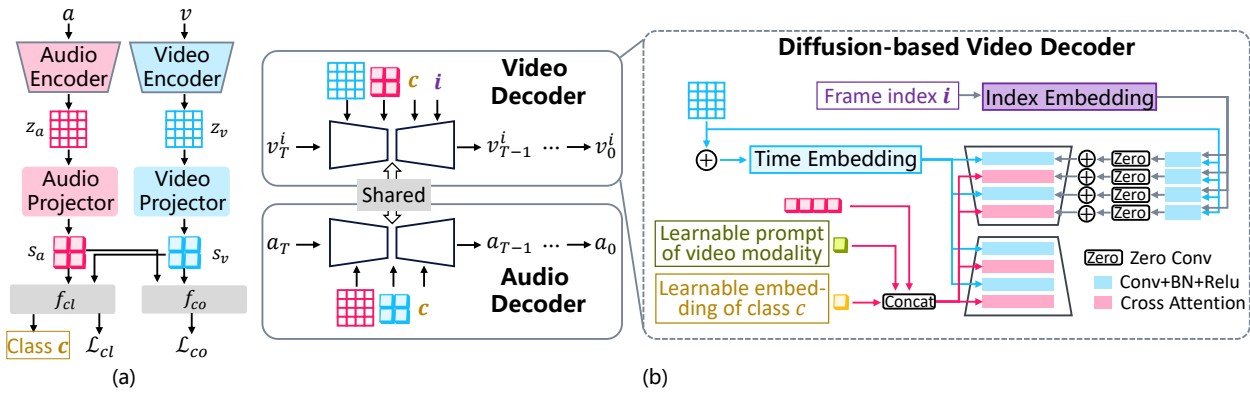

**Figure 2: Overall illustration of our multi-modal latent diffusion model (MM-LDM) framework. Modules with** gray **border comprise our hierarchical multi-modal autoencoder. The module with** orange **border is our transformer-based diffusion model that performs SVG in the latent space. The** green **rectangle depicts the modification of inputs for unconditional audio-video generation (i.e. SVG), audio-to-video generation, and video-to-audio generation, respectively.**

**Figure 3: The detailed architecture of our multi-modal autoencoder. (a) Given a pair of audio and video inputs, two modal-specific encoders learn their perceptual latents. Two projectors map from two respective perceptual latent space to the shared semantic space.** $\mathcal{L}_{cl}$ **represents the classification loss and** $\mathcal{L}_{co}$ **denotes the contrastive loss. Class information can be obtained from the classification head during inference. (b) We share the decoder parameters and incorporate multiple conditional information for signal decoding. For the video modality, we provide a specific input of frame index to extract information of the target video frame.**

the extracted features are of high-level semantic information, we employ a classification head $f_{cl}$ that takes a pair of audio and video features as inputs and predicts its class label, which is optimized using a classification cross-entropy loss. A contrastive loss is employed with a specified contrastive head to bridge the gap between video and audio features. The contrastive head $f_{co}$ maps $s_a$ and $s_v$ to 1D features respectively and calculates their contrastive loss with matched pairs of audio-video features being positive samples and all unmatched pairs of audio-video features as negative samples. Following [21], we define the contrastive loss as follows:

$$
\begin{aligned}
\mathcal{L}_{co} = &- \frac{1}{2} \sum_{i=1}^{B} \log \frac{\exp(\tau \cdot sim(f_{co}(s_a^i), f_{co}(s_v^i)))}{\sum_{j=1}^{B} \exp(\tau \cdot sim(f_{co}(s_a^i), f_{co}(s_v^j)))} \\
&- \frac{1}{2} \sum_{i=1}^{B} \log \frac{\exp(\tau \cdot sim(f_{co}(s_v^i), f_{co}(s_a^i)))}{\sum_{j=1}^{B} \exp(\tau \cdot sim(f_{co}(s_v^i), f_{co}(s_a^j)))}
\end{aligned}
\tag{1}
$$

where $sim(*)$ calculates the dot product of input features, $B$ and $\tau$ denote the batch size and a learnable parameter, respectively.

**Signal Decoding.** As illustrated in Fig. 3(b), during the reconstruction of video signals, the signal decoder takes into consideration multiple factors, including the video perceptual latent $z_v$, frame index $i$, audio semantic feature $s_a$, learnable modality embedding,

and learnable class embedding. The video perceptual latent feature plays a critical role by imposing strict constraints from two perspectives. First, for a dense spatial constraint, frame-specific multi-scale features are extracted from the latent feature using residual blocks and added to the outputs of the UNet encoder. These features encompass detailed spatial information for the $i$-th video frame. Second, to provide comprehensive global instruction, a content feature is derived by pooling the latent feature across spatial channels. This content feature is then added to the time embedding. The audio semantic features are rasterized and concatenated with the learnable modality embedding and the class embedding. This concatenated feature is then fed into cross-attention layers to provide rich conditional information. When dealing with audio reconstruction, the signal decoder employs similar inputs, excluding the frame index. More details can be found in the supplementary material.

**Training Targets.** Following [8], we utilize the $\epsilon$-prediction to optimize our signal decoder, which involves the noise mean square error loss $\mathcal{L}_{MSE}$. Following [23], we incorporate additional KL losses $\mathcal{L}_{KL}^a$ and $\mathcal{L}_{KL}^v$ to punish the distributions of audio and video latents towards an isotropic Gaussian distribution. Previous works have proven the effectiveness of adversarial loss in training single-modal autoencoders [4, 26]. Here, we introduce a novel adversarial loss to improve the quality of reconstructed multi-modal signals in

terms of both single-modal realism and multi-modal consistency. We first obtain a pair of decoded video frames $\langle \bar{v}_i, \bar{v}_j \rangle$ with $i < j$ and corresponding audio image $\bar{a}$. Then, for the optimization of the discriminator, we select $\langle a, v_i, v_j \rangle$ as the real sample and $\langle \bar{a}, \bar{v}_i, \bar{v}_j \rangle$, $\langle \bar{a}, \bar{v}_i, v_j \rangle$, $\langle \bar{a}, v_i, \bar{v}_j \rangle$, and $\langle \bar{a}, \bar{v}_j, \bar{v}_i \rangle$ to be the fake samples. $\langle \bar{a}, \bar{v}_i, \bar{v}_j \rangle$ is viewed as the real sample for our autoencoder. Our adversarial loss can be formulated as $\mathcal{L}_{GAN}^D$ for the discriminator and $\mathcal{L}_{GAN}^{AE}$ for our autoencoder:

$$
\begin{aligned}
\mathcal{L}_{GAN}^D =& log(1 - \mathcal{D}(\langle a, v_i, v_j \rangle)) + log\mathcal{D}(\langle \bar{a}, \bar{v}_i, \bar{v}_j \rangle) \\
&+ log\mathcal{D}(\langle \bar{a}, \bar{v}_i, v_j \rangle) + log\mathcal{D}(\langle \bar{a}, v_i, \bar{v}_j \rangle) \\
&+ log\mathcal{D}(\langle \bar{a}, \bar{v}_j, \bar{v}_i \rangle)
\end{aligned}
\tag{2}
$$

$$
\mathcal{L}_{GAN}^{AE} = log(1 - \mathcal{D}(\langle \bar{a}, \bar{v}_i, \bar{v}_j \rangle))
$$

Our discriminator is constructed by several spatio-temporal modules that consist of residual blocks and cross-modal full attentions. Our final training loss for the multi-modal autoencoder becomes:

$$
\begin{aligned}
\mathcal{L}_{AE} =& \mathcal{L}_{MSE} + \lambda_{cl}\mathcal{L}_{cl} + \lambda_{co}\mathcal{L}_{co} \\
&+ \lambda_{kl}(\mathcal{L}_{KL}^a + \mathcal{L}_{KL}^v) + \lambda_{gan}\mathcal{L}_{GAN}^{AE}
\end{aligned}
\tag{3}
$$

where $\lambda_{cl}$, $\lambda_{co}$, $\lambda_{kl}$ and $\lambda_{gan}$ are predefined hyper-parameters. $\mathcal{L}_{cl}$ and $\mathcal{L}_{co}$ are the classification and contrastive losses, respectively.

## 3.2 Multi-Modal Latent Diffusion Model

As illustrated in Fig. 2, our approach independently corrupts audio and video latents during the forward diffusion process, whereas in the reverse denoising diffusion process, we employ a unified model that jointly predicts noise for both modalities. In particular, during forward diffusion, we corrupt audio and video latents, which are denoted as $z_a^0$ and $z_v^0$ (i.e. $z_a$ and $z_v$), by $T$ steps using a shared transition kernel. For simplicity, we use $z^0$ to represent both $z_a^0$ and $z_v^0$ in the subsequent section. Following prior works [8, 24], we define the transition probabilities as follows:

$$
q(z^t|z^{t-1}) = \mathcal{N}(z^t; \sqrt{1 - \beta_t}z^{t-1}, \beta_t\mathbf{I})
\tag{4}
$$

$$
q(z^t|z^0) = \mathcal{N}(z^t; \sqrt{\bar{\alpha}_t}z^0, (1 - \bar{\alpha}_t)\mathbf{I})
\tag{5}
$$

where $\{\beta_t \in (0, 1)\}_{t=1}^T$ is a set of shared hyper-parameters, $\alpha_t = 1 - \beta_t$, and $\bar{\alpha}_t = \prod_{i=1}^t \alpha_i$. Utilizing Eq. (5), we obtain corrupted latents $z^t$ at time step $t$ as follows:

$$
z^t = \sqrt{\bar{\alpha}_t}z^0 + (1 - \bar{\alpha}_t)n^t
\tag{6}
$$

where $n^t \sim \mathcal{N}(0, \mathbf{I})$ represents noise features $n_a^t$ and $n_v^t$ for $z_a^t$ and $z_v^t$ respectively. The reverse diffusion processes of audio and video latents $q(z^{t-1}|z^t, z^0)$ have theoretically traceable distributions. To capture correlations between audio and video modalities, we introduce a unified denoising diffusion model $\theta$. This model takes both corrupted audio and video latents $(z_a^t, z_v^t)$ as input and jointly predicts their noise features. The reverse diffusion process of corrupted audio and video latents is formulated as:

$$
q((z_a^{t-1}, z_v^{t-1})|(z_a^t, z_v^t)) = \mathcal{N}((z_a^{t-1}, z_v^{t-1})|\mu_\theta(z_a^t, z_v^t), \tilde{\beta}_t\mathbf{I})
\tag{7}
$$

During training, we minimize the mean square error between the predicted and original noise features of matched audio-video pairs, known as $\epsilon$-prediction in the literature [11]:

$$
\mathcal{L}_\theta = \frac{1}{2}\|\tilde{n}_\theta^a(z_a^t, z_v^t, t) - n_a^t\|_2 + \frac{1}{2}\|\tilde{n}_\theta^v(z_a^t, z_v^t, t) - n_v^t\|_2
\tag{8}
$$

Here, $\tilde{n}_\theta^a(z_a^t, z_v^t, t)$ and $\tilde{n}_\theta^v(z_a^t, z_v^t, t)$ are the predicted audio and video noise features, respectively. Given that our audio and video latent features $z_a$ and $z_v$ possess relatively small spatial resolution, we employ a Transformer-based diffusion model known as DiT [20] as our backbone model. We rasterize audio and video latent features and independently add positional embeddings [28] to each latent. Then, two learnable token embeddings, $[EOS_a]$ and $[EOS_v]$, are defined and inserted before the audio and video features, respectively. Finally, audio and video latent features are concatenated and fed to DiT for multi-modal generation.

## 3.3 Conditional Generation

Inspired by the success of the classifier-free guidance [9, 22, 25], we employ a cross-modal sampling guidance that targets both audio-to-video and video-to-audio generation tasks. Our approach involves training the single MM-LDM to simultaneously learn three distributions: an unconditional distribution denoted as $\tilde{n}_\theta(z_a^t, z_v^t, t)$ and two conditional distributions represented as $\tilde{n}_\theta(z_v^t, t; z_a)$ and $\tilde{n}_\theta(z_a^t, t; z_v)$, corresponding to the SVG, audio-to-video and video-to-audio generation tasks respectively. To accomplish this, we incorporate a pair of null audio and video latents, defined as $(\mathbf{0}_a, \mathbf{0}_v)$ with $\mathbf{0}_a = 0$ and $\mathbf{0}_v = 0$. Then, the unconditional distribution $\tilde{n}_\theta(z_a^t, z_v^t, t)$ can be reformulated to be $\tilde{n}_\theta(z_a^t, z_v^t, t; \mathbf{0}_a, \mathbf{0}_v)$. The conditional distribution $\tilde{n}_\theta(z_v^t, t; z_a)$ can be reformed as $\tilde{n}_\theta(z_a^t, z_v^t, t; z_a, \mathbf{0}_v)$, where $z_a^t$ can be obtained directly given $z_a$ ant $t$ according to Eq. (6). Similarly, $\tilde{n}_\theta(z_a^t, t; z_v)$ is reformulated as $\tilde{n}_\theta(z_a^t, z_v^t, t; \mathbf{0}_a, z_v)$. As depicted in Fig. 2, the conditional inputs are added to the input latents after zero convolution layers (which are ignored in the figure for conciseness) to provide conditional information. We randomly select 5% training samples for each conditional generation task. Finally, taking audio-to-video generation as an example, we perform sampling utilizing the following linear combination of the conditional and unconditional noise predictions, defined as follows:

$$
\begin{aligned}
\bar{n}_\theta^v(z_v^t, t; z_a) =& \phi \cdot \tilde{n}_\theta^v(z_a^t, z_v^t, t; z_a, \mathbf{0}_v) \\
&- (1 - \phi) \cdot \tilde{n}_\theta^v(z_a^t, z_v^t, t; \mathbf{0}_a, \mathbf{0}_v)
\end{aligned}
\tag{9}
$$

where $\phi$ serves as a hyper-parameter that controls the intensity of the conditioning.

## 4 EXPERIMENT

## 4.1 Experimental Setups

**Dataset.** Following [24], we conduct experiments on three popular sounding video datasets, namely Landscape [15], AIST++ [16], and AudioSet [6]. Details of each dataset are presented in the supplementary material.

**Evaluation Metrics.** For video evaluation, we follow previous settings [5, 24, 35] that employ the Fréchet Video Distance (FVD) and Kernel Video Distance (KVD) metrics for video evaluation and Fréchet Audio Distance (FAD) for audio evaluation. Our MM-LDM synthesize all videos at a $256^2$ resolution. We resize the synthesized videos when testing the metrics in the $64^2$ resolution.

**Implementation Details.** When training our multi-modal autoencoder, we utilize pretrained KL-VAE [23] with the downsample factor being 8. Both video frames and audio images are resized to a $256^2$ resolution, and video clips have a fixed length of 16 frames

Table 1: Quantitaive comparison on tasks of 1) v: video generation, 2) a: audio generation, 3) svg: sounding video generation , 4) a2v: audio-to-video, and 5) v2a: video-to-audio generation. Results with ∗ are reproduced using released sources.

| Method | Resolution | Sampler | Landscape | | | AIST++ | | |
|---|---|---|---|---|---|---|---|---|
| | | | FVD ↓ | KVD ↓ | FAD ↓ | FVD ↓ | KVD ↓ | FAD ↓ |
| Ground Truth | $64^2$ | - | 16.3 | -0.015 | 7.7 | 6.8 | -0.015 | 8.4 |
| Ground Truth | $256^2$ | - | 22.4 | 0.128 | 7.7 | 11.5 | 0.043 | 8.4 |
| Single-Modal Video Generation | | | | | | | | |
| DIGAN [35] | $64^2$ | - | 305.4 | 19.6 | - | 119.5 | 35.8 | - |
| TATS-base [5] | $64^2$ | - | 600.3 | 51.5 | - | 267.2 | 41.6 | - |
| MM-Diffusion-v | $64^2$ | dpm-solver | 238.3 | 15.1 | - | 184.5 | 33.9 | - |
| MM-Diffusion-v* | $64^2$ | dpm-solver | 237.9 | 13.9 | - | 163.1 | 28.9 | - |
| MM-Diffusion-v+SR* | $64^2$ | dpm-solver+DDIM | 225.4 | 13.3 | - | 142.9 | 24.9 | - |
| MM-LDM-v | $64^2$ | DDIM | **122.1** | **6.4** | - | **83.1** | **13.1** | - |
| MM-Diffusion-v+SR* | $256^2$ | dpm-solver+DDIM | 347.9 | 27.8 | - | 225.1 | 51.9 | - |
| MM-LDM-v | $256^2$ | DDIM | **156.1** | **13.0** | - | **120.9** | **26.5** | - |
| Single-Modal Audio Generation | | | | | | | | |
| Diffwave [13] | - | - | - | - | 14.0 | - | - | 15.8 |
| MM-Diffusion-a | - | dpm-solver | - | - | 13.6 | - | - | 13.3 |
| MM-Diffusion-a* | - | dpm-solver | - | - | **9.6** | - | - | 12.6 |
| MM-LDM-a | - | DDIM | - | - | 10.7 | - | - | **11.7** |
| Multi-Modal Generation | | | | | | | | |
| MM-Diffusion-svg | $64^2$ | DDPM | 117.2 | 5.8 | 10.7 | 75.7 | 11.5 | 10.7 |
| MM-Diffusion-svg | $64^2$ | dpm-solver | 229.1 | 13.3 | 9.4 | 176.6 | 31.9 | 12.9 |
| MM-Diffusion-svg+SR* | $64^2$ | dpm-solver+DDIM | 211.2 | 12.6 | 9.9 | 137.4 | 24.2 | 12.3 |
| MM-LDM-a2v | $64^2$ | DDIM | 89.2 | 4.2 | - | 71.0 | 10.8 | - |
| MM-LDM-v2a | - | DDIM | - | - | 9.2 | - | - | 10.2 |
| MM-LDM-svg | $64^2$ | DDIM | **77.4** | **3.2** | **9.1** | **55.9** | **8.2** | 10.2 |
| MM-Diffusion-svg+SR* | $256^2$ | dpm-solver+DDIM | 332.1 | 26.6 | 9.9 | 219.6 | 49.1 | 12.3 |
| MM-LDM-a2v | $256^2$ | DDIM | 123.1 | 10.4 | - | 128.5 | 33.2 | - |
| MM-LDM-svg | $256^2$ | DDIM | **105.0** | **8.3** | **9.1** | **105.0** | **27.9** | 10.2 |

Table 2: Efficiency comparison with MM-Diffusion [24] on a V100 32G GPU, which models SVG within the signal space. MBS denotes the Maximum Batch Size.

| Method | HW | Training | | Inference | |
|---|---|---|---|---|---|
| | | MBS | Time/Step | MBS | Time/Sample |
| MM-Diffusion | $64^2$ | 4 | 1.70s | 32 | 33.1s |
| MM-Diffusion | $128^2$ | 1 | 2.36s | 16 | 90.0s |
| MM-LDM (ours) | $256^2$ | 9 | 0.46s | 4 | 70.7s |
| MM-LDM* (ours) | $256^2$ | **12** | **0.38s** | **33** | **8.7s** |

Table 3: Human Evaluation on the Landscape dataset.

| Method | AQ↑ | VQ↑ | A-V↑ |
|---|---|---|---|
| MM-Diffusion | 2.46 | 2.10 | 2.99 |
| MM-LDM | 2.98 | 3.68 | 3.29 |

($F = 16$). The audio and video encoders use downsample factors of $f_a = 4$ and $f_v = 2$, yielding latents of spatial size $8^2$ and $16^2$, respectively. The number of latent channels is 16 for both modalities. The loss weights $\lambda_{cl}$, $\lambda_{co}$, and $\lambda_{gan}$ are 1e-1, 1e-1, and 1e-1, respectively. The loss weight $\lambda_{kl}$ is set as 1e-9 for Landscape and 1e-8 for AIST++. Further details can be found in the supplementary material.

## 4.2 Quantitative Comparison

*Quality*. We quantitatively compare our method with prior works for single-modal generation (i.e. video or audio generation) and multi-modal audio-video joint generation (i.e. audio-to-video, video-to-audio, and sounding video generation) tasks. The results

are reported in Table. 1. For the single-modal generation tasks like audio generation and video generation, MM-Diffusion outperforms previous single-modal generation methods significantly on both datasets. However, on Landscape, our MM-LDM further outperforms MM-Diffusion by 103.3 FVD and 6.9 KVD at the $64^2$ resolution and 191.8 FVD and 14.8 KVD at $256^2$ resolution. On the AIST++ dataset, our MM-LDM outperforms MM-Diffusion by 59.8 FVD and 11.8 KVD at the $64^2$ resolution, 104.2 FVD and 25.4 KVD at the $256^2$ resolution, and 0.9 FAD. These results reveal the potential of our MM-LDM in modeling audio and video generation tasks.

For multi-model joint generation, we report the performance of MM-LDM on audio-to-video, video-to-audio, and sounding video generation tasks. It can be observed that MM-LDM synthesizes better video results with conditional audio inputs than non-conditions, and so is for audio generation. This phenomenon demonstrates that our MM-LDM can capture insightful cross-modal information to assist the generation of a single modality. For sounding video generation at the $64^2$ resolution, we achieve a 39.8 FVD, 2.6 KVD and 1.6 FAD improvement on the Landscape dataset and a 19.8 FVD, 3.3 KVD and 0.5 FAD improvement on the AIST++ dataset compared to MM-Diffusion. At the $256^2$ resolution, we achieve a 227.1 FVD, 18.3 KVD, and 0.8 FAD improvement on the Landscape dataset and a 114.6 FVD, 21.2 KVD and 2.1 FAD improvement on the AIST++ dataset. It can be seen that our method enhances the generation quality more significantly when the resolution increases, demonstrating the necessity of perceptual latent spaces for high-resolution

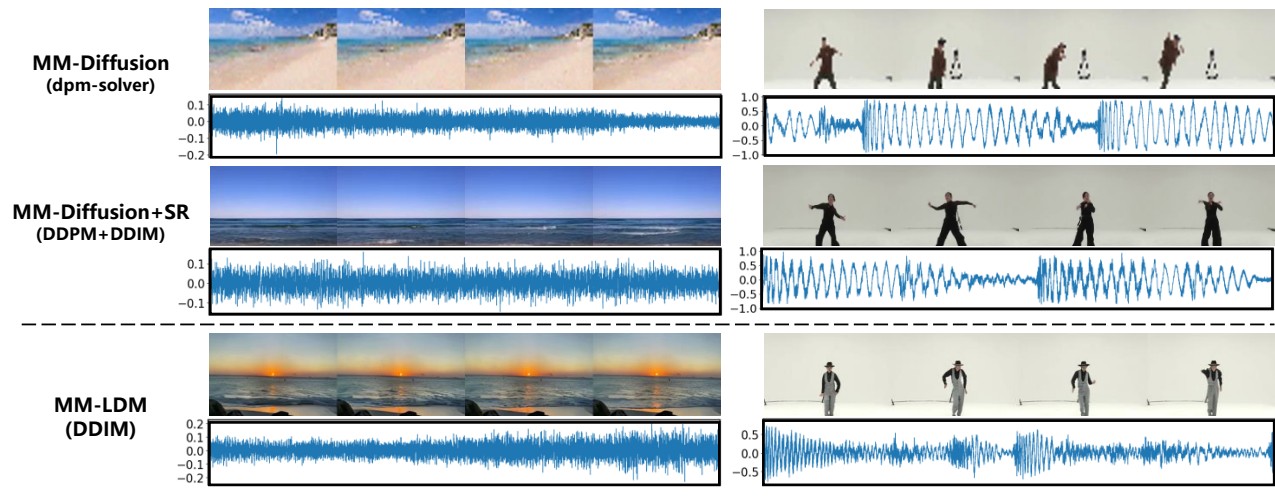

**Figure 4: Qualitative comparison of sounding video samples: MM-Diffusion vs. MM-LDM (ours). All presented audios can be played in Adobe Acrobat by clicking corresponding wave figures.**

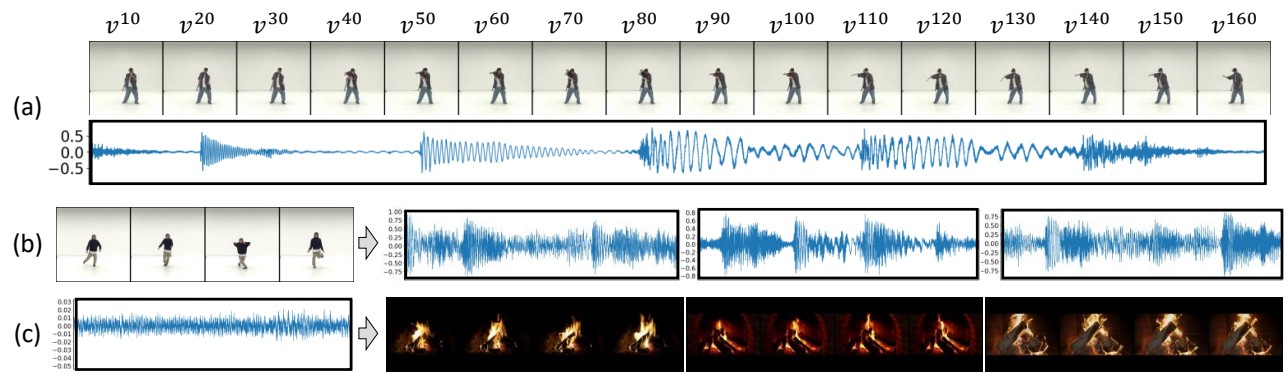

**Figure 5: Samples of (a) long sounding video generation, (b) video-to-audio generation, and (c) audio-to-video generation tasks.**

sounding video generation. Notably, when using the DDPM sampler, MM-Diffusion requires 1000 diffusion steps to synthesize a sounding video sample, taking approximately 8 minutes for a single sample. In contrast, our MM-LDM synthesizes higher-resolution videos with only 50 sampling steps using the DDIM sampler.

*Efficiency.* We quantitatively compare the efficiency of our MM-LDM and MM-Diffusion and present the results in Table. 2. MM-LDM incorporates both the auto-encoder and the DiT generator, while MM-LDM* only tests the DiT generation performance by pre-processing and saving all latents. We fix the batch size being 2 when testing the one-step time during training and the one sample time during inference, and DDIM sampler is used with 50 steps for both methods. Since MM-Diffusion operates in the signal space, it demands huge computational complexity when the spatial resolution of synthesized videos increases. In particular, it struggles to model high-resolution ($256^2$) video signals on a 32G V100, leading to the out-of-memory error. We evaluate the efficiency of MM-Diffusion with two spatial resolution settings: $64^2$ and $128^2$. MM-LDM demonstrates improved efficiency with higher video resolutions ($256^2$ vs.

$128^2$) during the training process. When employing the same batch size (i.e., 2), our MM-LDM outperforms MM-Diffusion by 6x speed for each training step, allowing a larger training batch size with a higher video resolution. During inference, our diffusion generative model DiT, which performs SVG within the latent space, achieves a 10x faster sampling speed and allows a larger sampling batch size.

## 4.3 Human Evaluation

For a thorough assessment, we conduct a manual evaluation of videos sampled by both MM-Diffusion and our MM-LDM on the landscape dataset. We randomly synthesize 500 sounding video samples using each model, obtaining a total of 1000 samples. These samples are then divided into five groups, with each group consisting of 200 samples for user rating. Following MM-Diffusion [24], each video is evaluated from three perspectives: audio quality (AQ), video quality (VQ), and audio-video matching (A-V), corresponding to sounding clarity, visual realism, and audio-video alignment, respectively. Each rating has a maximum score of 5. Each group of samples are shuffled and delivered to two users for voting. The average scores are shown in Table 3. Our MM-LDM significantly

**Table 4: Quantitative comparison with scaled-up data and model for open-domain generation.**

| Model | Params | FLOPS | FVD ↓ | KVD ↓ | FAD ↓ |
|-------|--------|-------|-------|-------|-------|
| MM-Diffusion [24] | 134M | 567G | 649.8 | 34.6 | 2.9 |
| MM-LDM-S | 131M | 34G | 185.8 | 10.1 | 1.59 |
| MM-LDM-B | 384M | 101G | 181.5 | 9.5 | 1.55 |
| MM-LDM-L | 1543M | 407G | 164.1 | 8.5 | 1.52 |

surpasses MM-Diffusion by 0.52 AQ, 1.58 VQ, and 0.30 A-V, demonstrating the effectiveness of our proposed method and the necessity of constructing multi-modal latent space.

## 4.4 Qualitative Comparison

We qualitatively compare the generative performance of our MM-LDM and MM-Diffusion in Fig. 4, using the provided checkpoints of MM-Diffusion when sampling. Videos synthesized by MM-Diffusion produce blurry appearances with deficient details, whereas our MM-LDM yields clearer samples with better audio-video alignments.

To further explore the adaptivity of our proposed method, we extend it to the long sounding video generation task in an autoregressive manner, which is specified in the supplementary material. We present the sample of long sounding video generation in Fig. 5(a), where our method can synthesize realistic and coherent long sounding videos with up to 160 frames. We also test the generation performance on conditional single-modal generation tasks like video-to-audio generation and audio-to-video generation tasks. As shown in Fig. 5(b) and Fig. 5(c), our method can generate consistent and diverse audio or video samples based on the video or audio condition. We also extend our method to other generation tasks like audio continuation and video continuation and provide samples in the supplementary material.

## 4.5 Generalization and Scaling Capability

To explore the generalization and scaling capability of our proposed MM-LDM, we conduct experiments using a larger open-domain dataset across various scales. Specifically, similar to MM-Diffusion, we filter 100K high-quality videos from the open-domain dataset AudioSet [6]. Then we optimize both MM-Diffusion and our MM-LDM on this dataset, testing the performance of MM-LDM at three different scales, i.e., small (S), base (B), and large (L). Each experiment is conducted on 8 A800 GPUs with comparable training times. As reported in Table 4, MM-LDM significantly outperformed MM-Diffusion. Furthermore, as the number of parameters of MM-LDM increases, the performance exhibits a significant boost, revealing the potential of our proposed method in terms of scaling capability. Samples of open-domain video generation are provided in the supplementary material.

## 4.6 Ablation Studies

We conduct ablation studies on our multi-modal autoencoder and report the results in Table. 5. Our base autoencoder independently decodes signals for each modality based on the respective spatial information from perceptual latents. To reduce computational complexity, we share the diffusion-based signal decoder for both modalities and initialize its parameters with [23]. Two learnable embeddings are incorporated to prompt each modality. For a more

**Table 5: Ablation study on the multi-modal autoencoder on the Landscape dataset.**

| Model | rFVD ↓ | rKVD ↓ | rFAD ↓ |
|-------|--------|--------|--------|
| MM-LDM | **53.9** | **2.4** | **8.9** |
| −Adversarial training loss | 75.7 | 3.9 | 8.9 |
| −Finetune KL-VAE decoder | 80.1 | 4.3 | 9.1 |
| −Semantic cross-modal feature | 87.7 | 5.1 | 9.1 |
| −Learnable class embedding | 94.4 | 5.7 | 9.2 |
| −Latent average pooling | 105.5 | 6.8 | 9.0 |
| −Learnable modality prompt | 110.7 | 6.9 | 9.3 |
| −Weight Initialization | 132.4 | 8.1 | 10.7 |

**Table 6: Sensitivity analysis on loss weights.**

| # Reso. | $\lambda_{kl}$ | $\lambda_{co}$ | $\lambda_{cl}$ | rFVD $64^2$ | rKVD $64^2$ | rFAD $64^2$ | rFVD $256^2$ | rKVD $256^2$ | rFAD $256^2$ |
|---------|------|------|------|------|------|------|------|------|------|
| 1 | 1e-8 | 1.0 | 1.0 | 68.0 | 11.2 | 10.0 | 89.7 | 23.8 | 10.0 |
| 2 | 1e-8 | 0.3 | 0.3 | 62.6 | 10.8 | 10.0 | 90.3 | 24.5 | 9.9 |
| 3 | 1e-8 | 0.1 | 0.1 | 59.6 | 10.4 | 9.8 | 85.1 | 21.4 | 9.9 |
| 4 | 1e-7 | 0.1 | 0.1 | 63.5 | 10.7 | 10.0 | 88.8 | 24.2 | 10.0 |

comprehensive content representation, we apply average pooling to each perceptual latent, adding the latent feature to the timestep embedding and further enhancing the model performance. By incorporating the classification and contrastive losses, we leverage prior knowledge of class labels and extract high-level cross-modal information, which significantly boosts model performance. Since the KL-VAE was originally trained on natural images and is unfamiliar with physical images like audio images, we finetune its decoder on each dataset for better performance. Finally, after training with the adversarial loss, our autoencoder attains its best reconstruction performance, achieving 53.9 rFVD, 2.4 rKVD, and 8.9 rFAD.

## 4.7 Sensitivity Analysis

Following prior work [27], we consistently utilized an adversarial learning weight of 0.1 in all experiments. In addition, we experimented with four configurations of loss weights, each obtained after a 20-epoch run (approximately 16K steps) on the AIST++ dataset. Across different settings, we assess metrics such as FVD, KVD, and FAD for samples at resolutions of 64 and 256. As reported in Table 6, where Reso. denotes resolution, the performance of configuration #3 outperforms that of other configurations at both $64^2$ and $256^2$ resolutions. Thus, we select configuration #3 as the default configuration in this paper.

## 5 CONCLUSION

This paper introduces MM-LDM, a novel latent diffusion model for the SVG task. To reduce the computational complexity, we establish a low-level perceptual latent space for each modality, which is perceptually equivalent to the raw signal space but encompasses a much smaller feature size. Moreover, we employ a dual safeguard mechanism to ensure cross-modal consistency between generated audio and video content. First, we derive a high-level semantic space to provide more insightful cross-modal information when decoding audio and video latents. Second, sufficient cross-modal communication is allowed during the generation progress by incorporating a full-attention mechanism into generative models. Our method achieves new state-of-the-art results on multiple benchmarks with improved efficiency and exciting adaptability.

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
