# OpenReview forum: "MM-LDM: Multi-Modal Latent Diffusion Model for Sounding Video Generation"
_acmmm.org/ACMMM/2024/Conference — MM2024 Poster_

### Official Review · Reviewer_oRoQ · 2024-05-24

**Rating:** 5
**Confidence:** 4

**Summary:**

The authors propose a novel multi-modal latent diffusion model (MM-LDM) to tackle the challenging task Sounding Video Generation (SVG). The proposed method first converts the audio and visual data into a unified representation, then applies a hierarchical multimodal autoencoder to construct a low-level unimodal latent space and a high-level semantic multimodal latent space. The proposed method is capable of unconditional audio-video joint generation as well as cross-modal conditional generation. Experiments on the Landscape and AIST++ datasets verify the effectiveness of the proposed method.

**Strengths:**

[+] Overall, this paper is a good extension of the pivotal work MM-Diffusion. The idea of leveraging a latent space for diffusion is reasonable and the hierarchical multimodal autoencoder and the unified multimodal representation warrant a stable and scalable audio-video joint generation.
[+] The task of audio-video joint generation (a.k.a. SVG) is quite challenging. While the proposed method achieves satisfactory generation results, the authors also extend its capability to cross-modal conditional generation tasks, showing its good generalization. Numerical results on two benchmarks also verify the effectiveness.
[+] The writing is good and easy to follow.

**Limitations:**

[-] Adopting a latent diffusion module for generation tasks is not that novel.
[-] From my point of view, I think one primary advantage of such a unified representation/token setting is the scalable capability. What will the method perform when training on a larger dataset or a larger model? I think the investigation of the scaling law of such unified generation methods would be interesting and valuable for the community.
[-] Can the model be used for zero-shot conditional generation? How does the model perform for videos longer than 1 second? Qualitative results or at least natural language descriptions could be provided if there is no available evaluation benchmark.

Overall I think this paper is good and is worthy to be published in the conference, yet addressing these limitations could make the paper more robust.

**Suitability:**

3

---

### Official Review · Reviewer_6dth · 2024-05-24

**Rating:** 3
**Confidence:** 3

**Summary:**

This paper proposes a novel multi-modal latent diffusion model (MM-LDM) for Sounding Video Generation (SVG). It employs an innovative hierarchical multi-modal autoencoder to transition from video and audio to a latent space, conserving computational resources while enhancing generation quality. However, the paper’s main contribution, which is the transformation of data to latent space, is somewhat limited as it has been previously validated by other works like LDM. Additionally, The superiority of a single multimodal VAE over two separate VAEs is unclear.

**Strengths:**

1. The overall paper is easy to follow.
2. Some motivations are easy to understand.
3. The figures displayed in the paper are helpful for readers to understand the proposed method.

**Limitations:**

1. Transforming data to latent space to improve performance has been validated by other work, such as LDM, so this paper's core contribution is somewhat limited.
2. The video in the attachment has no sound on my Windows 11 laptop, and it plays like a noise on my phone. I'm not sure if this is due to the quality of the video itself or if it requires a particular player for my phone or PC. It's hard to ascertain their alignment due to the noise.
3. It is not clear that one multimodal VAEs are superior to two separate VAEs. I question the feasibility of learning aligned multimodal features from such a limited dataset.

I am willing to raise my score if my concerns are addressed well.

**Suitability:**

3

---

### Official Review · Reviewer_5N6y · 2024-05-26

**Rating:** 3
**Confidence:** 3

**Summary:**

This paper presents MM-LDM, the first latent diffusion model for sound video generation. It first introduces a multi-modal autoencoder where the audio and video share the same semantic latent space to reduce the signal dimension. Afterward, a DiT-like model acts as a denoising network in the latent space. The quantitative evaluations show that the method outperform previous baseline and can support various applications.

**Strengths:**

1. The supplementary material is aboundant, which helps to understand the paper;
2. The paper is well-constructed and technically clear.

**Limitations:**

1. What is the architecture/framework of the autoencoder? Is it another diffusion model instead of the KL-VAE in the vanilla LDM? If so, how to initialize its parameters with the pre-trained VAE in LDM (L868)?
2. If the AE's decoder is another diffusion model, I think the proposed pipeline is closer to a cascade diffusion model instead of a latent diffusion model since the decoder also incorporates strong generative priors rather than performing low-level compression.
3. Since the autoencoder is the major contribution, it should have more analysis and ablation. For instance, is it possible to use separate KL-VAE encoders for each modality?

**Suitability:**

3

---

### Official Review · Reviewer_1vfS · 2024-05-27

**Rating:** 5
**Confidence:** 3

**Summary:**

The author explores the challenge of sounding video generating (SVG) in this paper and introduces a multi-modality video generation model called MM-LDM, operating within a latent space framework. The idea of incorporating the latent diffusion model into SVG task is interesting and the overall result is promising.

**Strengths:**

1. The task of SVG is interesting and the overall idea of modeling audio and video in latent space is promising.
2. The experimental results are promising which demonstrate the effectiveness of the proposed method.

**Limitations:**

1. Is there any performance analysis of the proposed multi-modal VAE and two separate VAEs?
2. Is it possible for the proposed method to generate longer video?

**Suitability:**

3

---

### Meta-Review · Area_Chair_C49s · 2024-07-01

**Recommendation:** Accept (Poster)
**Confidence:** 2

**Metareview:**

The AC goes through the paper, rebuttal and review comments. This paper got 4 reviews, including 2 weak accept, 1 borderline accept and 1 borderline reject. All reviewers acknowledge the motivation. However,  there are still several problems, such as the advantage of the joint VAE. I hope the authors can solve these problems in the camera-ready.